# Targeting Liver Metastases to Potentiate Immunotherapy in MS-Stable Colorectal Cancer—A Review of the Literature

**DOI:** 10.3390/cancers15215210

**Published:** 2023-10-30

**Authors:** Oran Zlotnik, Lucyna Krzywon, Jessica Bloom, Jennifer Kalil, Ikhtiyar Altubi, Anthoula Lazaris, Peter Metrakos

**Affiliations:** 1Cancer Research Program, Research Institute of the McGill University Health Centre, Montreal, QC H4A 3J1, Canada; oran.zlotnik@mail.mcgill.ca (O.Z.); lucyna.krzywon@mail.mcgill.ca (L.K.); jessica.bloom@mail.mcgill.ca (J.B.); jennifer.kalil@mail.mcgill.ca (J.K.); anthoula.lazaris@mail.mcgill.ca (A.L.); 2Division of General Surgery, McGill University, Montreal, QC H4A 3J1, Canada; ikhtiyar.altubi@mail.mcgill.ca

**Keywords:** immunotherapy, colorectal cancer, liver metastases, macrophages, CD8 T cells, MS stable, irradiation

## Abstract

**Simple Summary:**

Immunotherapy is an innovative treatment that is highly effective against certain cancers, such as skin and lung cancer. However, for colorectal cancer, one of the most prevalent cancers, it does not benefit most patients. Recent research suggests that by treating liver metastases first, immunotherapy might become effective for those with colorectal cancer. This review delves into the current evidence supporting this strategy, aiming to understand its underlying mechanisms and determine the best way to implement it. If this approach gains widespread acceptance, it could revolutionize how we treat colorectal cancer patients.

**Abstract:**

Immunotherapy has revolutionized the treatment of several cancers, including melanoma and lung cancer. However, for colorectal cancer, it is ineffective for 95% of patients with microsatellite-stable disease. Recent evidence suggests that the liver’s immune microenvironment plays a pivotal role in limiting the effectiveness of immunotherapy. There is also evidence to show that targeting liver metastases with locoregional therapies, such as surgery or irradiation, could potentiate immunotherapy for these patients. This review presents evidence from preclinical studies regarding the underlying mechanisms and from clinical studies that support this approach. Furthermore, we outline potential directions for future clinical trials. This innovative strategy could potentially establish immunotherapy as an effective treatment for MS-stable colorectal cancer patients, which are currently considered resistant.

## 1. Introduction

### 1.1. Immunotherapy Has Made a Revolution in Some Cancers but Not in Microsatellite-Stable (MS-S) Colorectal Cancer

Immunotherapy has transformed cancer treatment, affecting lung cancer [1,2], melanoma [3,4], hepatocellular carcinoma [5], and renal cell carcinoma [6]. However, its incorporation into the standard care for colorectal cancer, the third leading global cause of cancer-related death [7], remains ongoing. Only about 3% of rectal cancer patients [8], and an overall 5–15% of overall colorectal cancer patients, primarily those with microsatellite high disease (MSI-HIGH), respond favorably to immunotherapy [9,10,11]. Regrettably, the vast majority with MS-stable disease do not benefit from these treatments [9,10,11].

### 1.2. Immunotherapy Is Less Effective in the Presence of Liver Metastases

Even in cancers that traditionally respond well to immunotherapy, such as melanoma, lung cancer, and renal cancer, not all patients benefit from immunotherapy. Those with liver metastases tend to have a reduced response to immunotherapy, which is frequently associated with lower survival rates [6,12,13,14,15,16,17,18]. Immunotherapy, intended to “release the brakes” on immune-system activity, seems to be less effective when liver metastases are present. This potential immunoregulatory impact of liver metastases may stem from the interaction between liver cells and the cancer cells that form the metastases [19].

### 1.3. The Liver Is Known to Play an Important Role in Immune Regulation in Other Contexts

This interaction between the liver and the immune system is also evident in the use of immunosuppressive drugs—medications that inhibit immune responses. These drugs, which act as a regulatory brake on immune activity, are commonly used to prevent immune rejection in organ-transplant recipients. Notably, liver-transplant recipients typically require lower doses of immunosuppressive drugs compared to those receiving kidney transplants. This trend continues even in cases of combined liver and kidney transplants, where recipients require reduced immunosuppression compared with patients who underwent a kidney transplant alone [20]. This phenomenon provides additional evidence to support the role of the liver in systemic immunoregulation.

One compelling hypothesis for this immunoregulatory role of the liver is its function as a “gatekeeper”. Constantly exposed to blood flow from the intestines through the portal circulation, the liver encounters foreign antigens present in the portal blood. Robust immune responses to these antigens could potentially result in an uncontrolled ongoing immune response [21,22,23,24].

### 1.4. Potentiating Immunotherapy: The Role of Liver Metastases Directed Therapies

In recent years, there has been growing evidence suggesting that the elimination of liver metastases using liver-directed interventional therapies—such as surgery, radiation, and ablation—can potentiate immunotherapy.

This review will cover several key points. The first point to be addressed is the lessons derived from preclinical models. These models provide insights into the immunoregulatory pathways activated by the liver and reveal the mechanisms through which metastatic cancers exploit these pathways to evade immune surveillance.

The second topic to be discussed in this review pertains to insights from clinical trials involving immunotherapy in MS-stable colorectal cancer. These clinical studies offer an overview of the response rates to immunotherapy in MS-stable colorectal cancer patients. Additionally, they shed light on clinical scenarios and subpopulations of MS-stable colorectal cancer patients who may benefit from immunotherapy.

Moving on to the third point of discussion in this review, we will delve into lessons learned from clinical studies that combine immunotherapy with liver-directed locoregional therapies. The objective here is to target liver metastases with locoregional therapies in order to limit the liver metastases’ ability to suppress the immune response against extrahepatic tumors, thus potentially enhancing the efficacy of immunotherapy.

Finally, this paper will address the potential for future clinical trials designed in accordance with the principles elucidated by the preclinical and clinical studies outlined in this review. These proposed clinical trials will involve the addition of immunotherapy to the standard-of-care therapy. This approach will enable the assessment of immunotherapy efficacy after the elimination of liver metastases, all without compromising the standard-of-care therapy.

Overall, this review aims to examine and evaluate key studies that explore liver-targeted interventional therapy as a strategic approach to enhance the effectiveness of immunotherapy. This approach holds promise as a potential therapeutic option for colorectal cancer patients with MS-stable disease, which constitutes the majority of patients.

## 2. Enhancing Immunotherapy Efficacy by Targeting Liver Metastases: Lessons from Preclinical Models

### 2.1. The Role of Myeloid-Derived Suppressor Cells (MDSCs) and Their Effect on CD8 Cells in Animals with Liver Tumors

Yu et al. [25] sought to understand the diminished response to immunotherapy observed in the presence of active liver metastases. To this end, they established a dual tumor mouse model, incorporating liver metastases from murine colorectal cancer (MC38 cell line) alongside a corresponding subcutaneous tumor.

In this mouse model, the immunotherapeutic response varied depending on the presence or absence of liver tumors. Mice bearing only MC38 subcutaneous tumors showed a favorable response to immunotherapy. In contrast, those with both MC38 subcutaneous and MC38 liver tumors did not exhibit such a response to immunotherapy. Moreover, when mice had a colorectal subcutaneous tumor (MC38) and melanoma liver tumors (B16F10—murine melanoma), their response to immunotherapy was favorable, mirroring the response in mice without liver tumors. This pattern indicates that the diminished immunotherapeutic response is specifically linked to the presence of similar antigens in both liver and extrahepatic tumors.

Further observations in this model revealed that mice with liver tumors had a systemic reduction in antigen-specific T cells. Building on this, the same research team found that human patients with liver metastases from prostate, breast, and colorectal cancers displayed decreased intratumoral CD8 effector-cell activity. Delving deeper into the mouse model with OT-I cells, it was determined that CD8 cells undergo apoptosis in liver tumors, a process likely triggered by CD11b+ myeloid cells.

Interestingly, this reduced immunotherapeutic response could be reversed. When liver-directed irradiation was applied in the mouse model, there was a notable decrease in CD11b+F4/80+ cells and an increase in T-cell infiltration within the liver. This liver irradiation not only enhanced the efficacy of immunotherapy but also prevented the loss of T cells.

### 2.2. Tumor-Infiltrating T Regulatory Cells in Mice Models with Liver Tumors

A different research group employed a similar mouse model with synchronous subcutaneous and liver tumors [26]. This group similarly found that mice with liver tumors exhibited a diminished response to immunotherapy. However, mice with subcutaneous tumors and synchronous tumors in nonhepatic locations (such as the peritoneum, lungs, or kidneys) responded well to immunotherapy. Notably, these variations in tumor response were absent in immunodeficient mice (SCID), suggesting that the differences are tied to the functionality of the adaptive immune system.

Upon analyzing the T cells infiltrating the subcutaneous tumors, researchers found that CD8 cells in mice with liver tumors showed decreased levels of CTLA-4, PD-1, ICOS, and KI67. Conversely, T regulatory cells (CD4 FOXP3) in these mice displayed increased expression of CTLA-4, ICOS, and PD-L1. Subsequent investigations revealed that the diminished activity of CD8 cells was primarily attributed to a subpopulation of tumor antigen-specific CD8 cells. Depletion of T regulatory cells resulted in the restoration of activity in tumor-specific CD8 cells within subcutaneous tumors.

It was found that the behavior of T regulatory cells impacted the function of MDSCs (myeloid-derived suppressor cells). Earlier work by Yu et al. [25] has established that MDSCs were potent suppressors of tumor-specific immunity. This observation is consistent with prior studies highlighting the suppressive role of MDSCs in tumor immunity [27,28,29,30].

## 3. The Potential of Immunotherapy in MS-Stable Colorectal Cancer Patients—Insights from Clinical Trials

Several clinical trials have utilized immune-checkpoint inhibitors to treat patients with MS-stable metastatic colorectal cancer (see Table 1). Specifically, the REGONIVO trial [31] utilized a combination of Nivolumab (a PD-1 inhibitor) and Regorafenib (a multikinase inhibitor) in patients with metastatic colorectal and gastric cancer who had not responded to two prior chemotherapy lines. In the colorectal cancer patients group, 50% of the patients without liver metastases (*n* = 8/16) showed a complete or partial response to the treatment, while only 15% with liver metastases (*n* = 2/13) had a similar response pattern. Responses were evaluated by the investigators per the Response Evaluation Criteria in Solid Tumors (RECIST). Gastric cancer patients exhibited a similar trend: a 40% response rate with liver metastases and a 67% response rate without liver involvement. The study evaluated the PD-L1 combined positive score (CPS), which is calculated by determining the percentage of tumor cells that are PD-L1 positive. However, this trial did not find a distinct correlation between CPS and treatment outcomes.

Notably, earlier studies that paired immune-checkpoint inhibitors (ICI) with bevacizumab for a similar group of patients did not report significant treatment benefits [41,42]

While both Regorafenib and Bevacizumab target the VEGF pathway, Regorafenib also interacts with other molecules, such as CSF1R. This molecule is predominantly found on myeloid cells and influences the creation and differentiation of macrophages. Mouse studies [25], have shown that inhibiting CSF1R can potentiate immunotherapy for colorectal liver metastases. This might provide insight into why the combination of Regorafenib and ICI yielded better outcomes than the ICI and Bevacizumab pairing.

Several additional recent studies reinforce the idea that colorectal cancer patients could benefit from immunotherapy irrespective of their MSI status [33,35,43]. Notably, the Niche trial [33] offers compelling evidence for the potential effectiveness of immunotherapy in MS-stable colorectal tumors without liver metastases. In this study, early-stage primary colorectal tumor patients received immunotherapy before surgical removal of the tumor. Among the 17 MS-stable colorectal cancer patients in the trial, 27% responded to immunotherapy, with post-treatment evaluations revealing 50–100% of their tumor cells as nonviable. Additionally, four patients showed partial responses, with post-treatment viable tumor cells ranging from 60 to 85%. These findings further suggest that immunotherapy could be advantageous for MS-stable colorectal cancer patients, especially those without liver metastases.

Recent findings from the MEDITREME trial [35] shed additional light on a subset of MS-stable colorectal cancer patients who might benefit from immune-checkpoint inhibitors (ICI). This study assessed the combined use of Dorvalumab and Tremelimumab, with standard-of-care chemotherapy in 57 patients with RAS-mutated, unresectable stage 4 colorectal cancer. The trial reported a progression-free survival of 8.2 months, surpassing the expected 5–6 months, especially noteworthy given that RAS-mutated cancers typically have a worse prognosis. The dual immunotherapy approach (Dorvalumab + Tremelimumab) in the trial achieved an objective response rate of 63%, which is notably higher than the 36% observed in the BECOME trial [44] that used a single immunotherapy. The study also highlighted that patients with high CD8 cell infiltration and elevated PD-L1 expression were more likely to respond positively to immunotherapy.

## 4. Combining Immunotherapy with Liver-Targeted Loco-Regional Interventions

### 4.1. Immunotherapy Combined with Liver Resection

Wang et al. carried out a retrospective analysis examining the effects of immunotherapy on MS-stable colorectal cancer patients, specifically focusing on the presence or absence of liver metastases [34]. The study encompassed 95 metastatic microsatellite-stable colorectal cancer patients who did not respond to two prior chemotherapy treatments. These individuals were subsequently treated with immune-checkpoint inhibitors (either anti-PD-1 or anti-PD-L1). Within this cohort, some patients were treated with immunotherapy after their liver metastases were surgically removed, some had ongoing liver metastases during their course of immunotherapy, and a third subset never developed liver metastases.

The most favorable immunotherapy outcomes were observed in patients without any history of liver metastases. These patients exhibited the longest survival rates. Conversely, those with ongoing liver metastases during their immunotherapy treatment showed the least favorable outcomes. Notably, patients who received immunotherapy after surgical removal of their liver metastases demonstrated better response rates than those with ongoing liver metastases.

The disease control rate stood at 58.5% for patients without liver metastases, in stark contrast to a mere 1.9% for those with liver metastases. The median duration of progression-free survival during immunotherapy was 3.0 months for patients postliver resection, compared to 1.5 months for those with active liver metastases. On multivariate analysis, which considered factors like age, gender, primary tumor location, ECOG status, and various mutation statuses, the presence of liver metastases during treatment emerged as the most influential factor affecting progression-free survival during immunotherapy. This association was statistically significant, with a hazard ratio of 7.0 (*p* < 0.01).

In a related study, Kanikarla et al. [36] utilized a dual-agent immunotherapy approach in the preoperative setting to treat 24 patients with colorectal liver metastases, aiming to explore the mechanisms of immunotherapy resistance in colorectal cancer prior to liver-metastases resection.

Liver metastases were analyzed for immune cell infiltration and molecular features. A complete response to immune-checkpoint inhibitors (ICI) was observed only in patients with high microsatellite instability (MSI-HIGH) disease. A lack of response was seen in patients with low microsatellite instability (MS-Stable) disease, which was expected.

However, immune profiling revealed that even patients with microsatellite-stable disease exhibited some immune response to ICI. Post-treatment flow cytometry showed an increase in the expression of LAG3, PD1, ICOS, and TIM3. Additionally, CD-8 cells were more frequent in post-treatment multiplex immunofluorescence, while T regulatory cells and macrophages decreased. This decrease in macrophage activity, as well as increases in CD86 (an antigen-presenting cell-specific marker) and CD69 (a marker of early activation), align with findings in mouse models [24,25], which suggest that liver-residing macrophages are inducing the apoptosis of tumor-specific CD8 cells. Moreover, a trend towards a lower frequency of CD8+ PD1+ cells was observed in patients with shorter disease-free survival. However, MS-stable disease with preoperative immunotherapy did not exhibit a clinical response. This might be explained by the sequence of treatment. As previously discussed, liver metastases appear to limit the efficacy of immunotherapy. Therefore, resecting the liver metastases before administering checkpoint inhibitors might have resulted in a measurable clinical response to immunotherapy, consistent with the findings from the study by Wang et al. [34].

### 4.2. Immunotherapy Combined with Irradiation and Radioembolization of Liver Metastases

Another modality that may be used to eliminate liver metastases in the context of potentiating immunotherapy is radiation therapy, similar to that previously described in animal models [37,38,39]. Parikh et al. conducted an open-label phase II trial where they combined radiation with ipilimumab and nivolumab to treat patients with metastatic MSS CRC (N = 40) [37]. In this trial, patients were first treated with Nivolumab (a PD-1 inhibitor) and Ipilimumab (an anti-CTLA-4 agent). Following the initial dose of immunotherapy, patients received a second dose of both drugs two weeks later and then began radiation therapy (a total of 24 Gy). Tumor samples were analyzed using whole exome sequencing, and blood samples were taken for DNA germline control. All participants were confirmed to have MS-stable tumors, with the majority presenting with metastatic disease. Prior to treatment, only 1 out of 40 patients had stable disease. Post-treatment, 25% of the patients treated per the protocol achieved stable disease, and 10% showed an objective response based on an intention-to-treat analysis. These findings suggest that targeting liver metastases with radiation might enhance the efficacy of immunotherapy.

Monjazeb et al. conducted a multicenter randomized trial where patients with MS-stable colorectal cancer were assigned to one of two radiation protocols combined with immunotherapy [38]. The primary goal was to observe a response outside the radiation field. Eighteen patients, who had undergone a median of four previous therapy lines, participated. However, in this trial, the clinical benefits were minimal, with only one patient showing a response outside the irradiated field. However, the study did find treatment-associated changes in the T-cell repertoire within the tumor and blood, and an increased T-cell infiltration in the irradiated area. It is important to note that, in this study, only one liver lesion was irradiated, and not necessarily all liver lesions, which may have led to a reduced response, as either viable liver metastasis could have played a role in restraining the response to immunotherapy.

In another clinical trial led by Floudas et al. [39], a combination of a PD-1 inhibitor (amp224), cyclophosphamide, and stereotactic body radiation was used. Out of the 15 patients in the trial, three experienced stable disease, but no complete or partial responses were recorded. Despite the limited clinical benefits observed, the study found that the polarization of M2 macrophages in the tumor microenvironment was reversed, aligning with findings from animal studies.

Lastly, a phase 2 study involved nine patients with MS-stable colorectal cancer liver metastases [40]. They were treated with Y-90 radioembolization, followed by a dual-agent immunotherapy regimen—Tremelimumab and Durvalumab. Unfortunately, all the patients experienced disease progression during or after their first two treatment cycles. The tumor’s lymphocyte infiltration was found to be low, and only transient gene expressions related to radiation treatment were identified.

Overall, it appears that when radiation therapy is combined with immunotherapy, the outcomes vary in treating MS-stable colorectal cancer patients. Some studies suggest potential benefits, while others indicate limited clinical efficacy. These differences might be attributed to the proportion of liver metastases treated in these studies, as well as the sequence of treatment or irradiation protocol. Further research is essential to ascertain the potential of this combination.

## 5. Directions for Potential Clinical Trials

### 5.1. A Window of Opportunity Trial

Many colorectal cancer patients are initially diagnosed with both a primary colorectal tumor and synchronous liver metastases [45,46]. Leading medical institutions often prioritize the removal of these liver metastases as the initial step in the treatment sequence. Following this, patients typically undergo chemotherapy and then have the primary colorectal tumor resected in a subsequent procedure [47,48].

The period between the liver metastases resection and the primary colorectal tumor resection provides an opportunity to investigate if an MS-stable primary tumor becomes more responsive to immunotherapy after liver-metastasis removal. By administering immunotherapy with standard chemotherapy after liver-metastasis resection but before primary tumor removal, we can evaluate the response through imaging and histological assessments during the primary tumor resection. A prospective trial, such as this, could corroborate Wang et al.’s findings [34], which showed patients with MS-stable disease responding to immunotherapy post-liver-metastasis resection. This is also consistent with the NICHE study [33] where 27% of patients with MS-stable early rectal cancer exhibited a significant response. This approach can be framed as a “window of opportunity” trial, adding immunotherapy to the standard treatment, to test if MS-stable colorectal cancer shows enhanced responsiveness to immunotherapy post-liver-metastasis removal (See Figure 1).

### 5.2. Measuring the Effect of Immunotherapy on Extrahepatic Metastases following the Resection of Liver Metastases

While not a standard of care, clinical evidence suggests that resecting liver metastases, even in the presence of extra-hepatic metastases, may provide significant clinical benefits. This includes extended survival compared to palliative chemotherapy. As an illustration, Darren et al. documented a five-year survival rate of 27% [49], while Elias et al. recorded a rate of 28% [50]. It is essential to note that these liver resections are likely suitable for a specific subset of patients and have a high probability of recurrence. However, introducing immunotherapy after liver resection could enhance its efficacy against extrahepatic diseases, such as lung or lymph node metastases. This is in line with the results from animal studies [25] and human retrospective studies [34]. Such a strategy could be explored in a clinical trial where patients undergo liver resection followed by immunotherapy, enabling the assessment of immunotherapy’s impact on extrahepatic metastases, including those in the lungs, lymph nodes, and peritoneal region (see Figure 1).

## 6. Summary

This review delves into various aspects of the recent literature regarding the potential for targeting liver lesions to enhance the effectiveness of immunotherapy in MSI-stable colorectal cancer, traditionally considered unresponsive to such treatment.

The preclinical findings outlined in this review shed light on the mechanisms by which liver metastases induce apoptosis in CD8 T cytotoxic cells. T cytotoxic cells play a pivotal role as the most potent effectors in the anticancer response triggered by immunotherapy. Their apoptosis effectively hinders the impact of immunotherapy. Furthermore, T regulatory cells, which inhibit T cytotoxic cells, seem to be more prevalent in extrahepatic tumors when experimental animals have liver metastases.

These findings resonate with a well-established pattern observed in cancers like melanoma or lung cancer, which are known for their favorable response to immunotherapy. Patients with liver metastases in these cancers exhibit significantly reduced response rates, further emphasizing the impact of liver metastases on the efficacy of immunotherapy.

Contrary to the prevailing paradigm, some subpopulations of MSI-stable colorectal cancer patients appear to be responsive to immunotherapy. These include patients with early rectal cancers, as described in the NICHE trial [33], who have not yet developed liver metastases. Moreover, it seems that resecting liver metastases can potentiate immunotherapy in human patients with MSI-stable colorectal cancer [34], aligning with the results observed in preclinical models of colorectal cancer. However, this effect was not as evident with other locoregional modalities, such as ablation or irradiation.

The findings and results presented in this review should be approached with caution, as they are susceptible to several biases. First, some of the studies included in this review, such as the study by Wang et al. [34], are retrospective studies, which makes their results prone to patient selection bias, misclassification bias, and variability in the type of immunosuppression prescribed to patients. Secondly, industry involvement in clinical trials, particularly those involving the use of immunotherapy, could have influenced the reported results. This also holds true for trials that require highly specialized equipment, such as microwave ablation generators and applicators. Lastly, the range of results may have been influenced by underreporting negative results, especially in retrospective studies.

It is crucial to recognize that the potential strategy described in this review may have several limitations. First, the potential of immunotherapy to benefit patients following the resection of liver metastases is limited, as the response to immunotherapy ranges between 30 and 60% in studies reporting the highest response rates [31,33,34] and is, therefore, probably only relevant for a subgroup of patients yet to be identified. Secondly, no prospective trial has reported the efficacy of immunotherapy following the resection of liver metastases, and, lastly, there is no known equivalent of this strategy in cancer types that are responsive to immunotherapy, such as melanoma or lung cancer.

Nevertheless, after taking all the caveats into account, this remains a promising strategy to enhance immunotherapy for patients who currently do not respond to it, potentially benefiting from its therapeutic value. This is especially relevant in the clinical scenarios described in our suggestions for future clinical trials, such as patients with locally advanced rectal cancers and liver metastases or patients with stage 4 disease and minimal response to previous lines of chemotherapy.

## 7. Conclusions

In conclusion, the accumulating evidence suggests that certain patients with colorectal liver metastases may benefit from immunotherapy. The elimination of liver metastases, whether through surgery or other forms of locoregional therapies, emerges as a promising avenue to amplify the response to immunotherapy, even in cases of advanced metastatic cancer. By strategically targeting liver metastases, we not only pave the way for treating extrahepatic disease with immunotherapy but could also offer a potential strategy to reduce the size of locally advanced primary tumors. This approach is particularly promising, given that the resection of these tumors can lead to considerable morbidity and mortality. However, proof of concept will be required in well-designed prospective clinical trials.

## Figures and Tables

**Figure 1 cancers-15-05210-f001:**
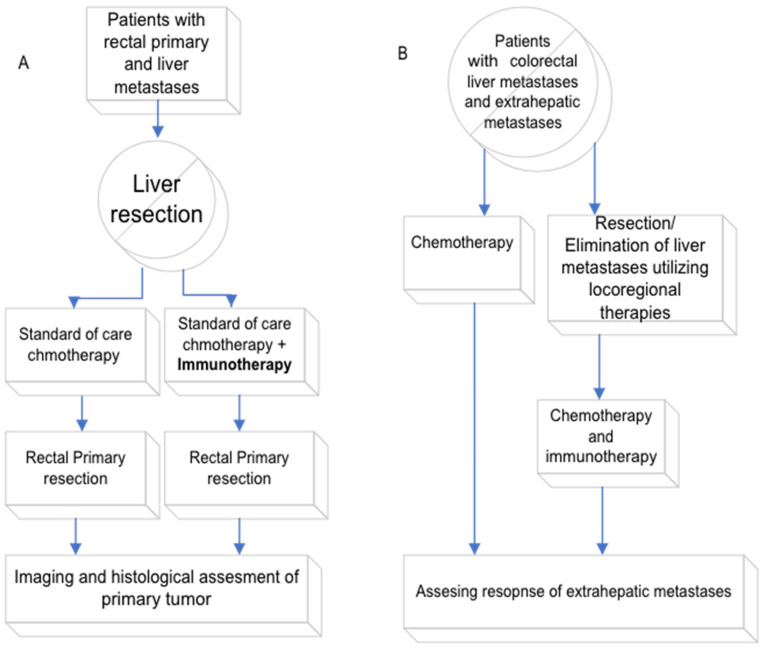
Potential future clinical trials. The figure illustrates potential future clinical trials exploring the use of liver-metastases resection to enhance immunotherapy for MS-stable colorectal cancer. (**A**) The trial involves resection of liver metastases as a “liver first” approach. This is followed by immunotherapy, alongside standard-of-care chemotherapy. A control group will receive chemotherapy alone postliver resection. Following these treatments, resection of the rectal primary tumor will take place. Treatment efficacy will be evaluated through imaging and histological analysis of the primary rectal tumor. (**B**) This potential trial focuses on leveraging liver-metastases resection to boost immunotherapy’s effectiveness against extrahepatic metastases. Participants will undergo liver-metastases resection, followed by a combination of immunotherapy and standard-of-care chemotherapy. This regimen will be compared to chemotherapy alone, which is the standard of care for patients with metastases across multiple sites. Imaging will be utilized to assess the response to the treatment.

**Table 1 cancers-15-05210-t001:** Summary of included Clinical Studies Utilizing Various Strategies to Potentiate Immunotherapy for MS-Stable Colorectal Cancer.

Year	Author	Study Type	Patient Cohort	Intervention	Key Findings
2019	Fukuoka et al. [31]	Prospective	Patients with metastatic MS-stable CRC, with no response to 2 previous lines of chemotherapy	Regorafenib + Nivolumab	50% response rate without liver metastases vs. 15% with liver metastases
2022	Mettu, N.B., et al. [32]	Prospective	Patients with mCRC who progressed on multiple therapies	Capecitabine + Bevacizumab + Immunotherapy	23% response rate in patients without liver metastases vs. 4.8% in patients with liver metastases
2020	Chalabi, M., et al. [33]	Prospective	Early stage primary rectal tumors	Immunotherapy before surgery	27% response rate in early-stage primary tumors
2021	Wang et al. [34]	Retrospective	Patients post failure of two previous chemotherapy lines	Comparing patients with liver metastases to patients with absent/resected liver metastases	Disease control rate of 58% without liver metastases vs. 1.9% with active liver metastases
2023	Thibaudin, M., et al. [35]	Prospective	RAS-mutated unresectable stage 4 CRC	Dual immunotherapy + modified FOLFOX	PFS of 8.2 months, surpassing the expected 5–6 months. ORR of 63% vs. 36% in previous studies
2021	Kanikarla et al. [36]	Prospective	Patients with colorectal liver metastases	Dual-agent immunotherapy before surgery	Lack of clinical response in patients with low microsatellite instability (MS-stable) disease
2021	Parikh et al. [37]	Prospective	Patients with metastatic MS-stable CRC	Radiation and dual-agent immunotherapy	Disease control rate of 37% in patients treated per protocol
2021	Monjazeb [38]	Prospective	Patients with advanced CRC and multiple previous chemotherapy lines	Radiation and dual-agent Immunotherapy	Only 1/18 patients showed a response outside the irradiated field, but immunotherapy-related changes observed in T cell repertoire in tumor and blood.
2019	Floudas [39]	Prospective	Metastatic colorectal cancer, refractory to standard chemotherapy	Pd-1 inhibitor combined with irradiation	There were no complete or partial responses, 3/15 patients in the trial experienced stable disease
2020	Wang [40]	Prospective	Patients with MSS metastatic CRC with liver-predominant disease who progressed following at least one prior line of treatment	Y-90 radioembolization followed by dual-agent immunotherapy	No response observed, study closed after 9 patients due to futility

Summary of clinical trials discussed in this review, utilizing different strategies to potentiate immunotherapy for MS-stable colorectal cancer.

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
