# Peer review of "Targeting Liver Metastases to Potentiate Immunotherapy in MS-Stable Colorectal Cancer—A Review of the Literature"

_cancers, 2023, doi:10.3390/cancers15215210_

Round 1

Reviewer 1 Report

Comments and Suggestions for Authors

Oran et al. wrote a timely and important manuscript on the role of liver metastases and liver-directed therapies in improving the effectiveness of immunotherapy in colorectal liver metastases. The manuscript comprehensively reviews preclinical and clinical evidence to support the hypothesis that liver-directed interventions can enhance immunotherapy outcomes, particularly in microsatellite-stable (MSS) colorectal cancer patients. The review is well-structured, with detailed explanations of the mechanisms and findings from various studies. However, there are areas for improvement:

Clarity and Structure: The manuscript is well-structured, but some sections could be further improved for clarity. The introduction section, in particular, could be more concise.

Bias and Limitations: The manuscript should address potential biases and limitations in the reviewed studies. For example, discussing the possible influence of industry funding or publication bias in the clinical trials could provide a more balanced perspective. Additionally, causality cannot be determined in the non-randomized studies.

Conclusion: The conclusion is somewhat abrupt. A more comprehensive summary of the main findings and their implications for future research and clinical practice would enhance the conclusion section.

Supporting Figures and Tables: Including figures or tables summarizing key findings from clinical trials or preclinical studies (e.g., response rates) would enhance the visual appeal and clarity of the manuscript.

Specific Feedback:

Introduction, paragraph 1.1: Specify the different percentages of MSI-high patients in both colon and rectal cancer.

Paragraph 3: The authors use the terms "favourable response" or "responded to," which are subjective terms that should be avoided. It is expected to use the RECIST criteria.

In paragraph 4.1, the authors use the term "survival benefit." Such a term should be reserved for randomized prospective trials. Causality cannot be determined in a retrospective fashion.

Typo: paragraph 4.2 is mistakenly named "4.1."

Overall, the manuscript provides valuable insights into the potential of liver-directed therapies to enhance immunotherapy outcomes in colorectal cancer. By addressing the areas for improvement, the manuscript can be even more impactful and informative for the readers.

Author Response

Thank you very much for your insightful comments.

We have carefully addressed your comments and recommendations in the following ways:
- The introduction section has been revised as per your suggestions.
- We have tackled the limitations and biases, paying special attention to industry influence and publication biases.
- The conclusion section has not only been edited but also expanded to encompass a discussion on the key findings from the studies in question.
- We have incorporated the percentages of MSI-HIGH disease specific to rectal and colorectal cancer within the introduction.
- The description of the response to treatment now includes references to the RECIST criteria.
- Adjustments have been made to the term "survival benefit," and we have ensured the correction of paragraph numbering.

Your time and comprehensive feedback are greatly appreciated. Thank you once again.

Reviewer 2 Report

Comments and Suggestions for Authors

The background description could clearly state the issues to be discussed and the reasons involved. First, the authors indicated that immunotherapy can be more effective when targeting liver metastases in preclinical models. Besides, they presented that immunotherapy could be a valuable option for MS-stable colorectal cancer patients by reviewing literatures of clinical trials. These findings rationally suggested targeting liver metastases a significant breakthrough in patients with MS-stable colorectal cancer. The authors also present some examples of immunotherapy combined with liver-targeted local intervention and provide some directions for potential clinical trials. This review article was written in an understandable scientific language and was found suitable for publication in this journal. However, some discussion/ summary about weakness and limitation about this strategy by targeting liver metastases in MS-stable colorectal cancer patients are recommended. Besides, minor errors in article format need to be corrected.

Reviewer 3 Report

Comments and Suggestions for Authors

Dear Authors, although the topic of the review is interesting, I find it written quickly. I propose a small revision on formatting (missing or present spaces, repeated paragraphs 1.3, 4.1...cmbining instead of combining...). I also propose to write in full the acronyms used, already in abstract, such as MS, MSI because you must always facilitate the reader, do not discourage him.
